# Towards a Responsible Transition to Learning Healthcare Systems in Precision Medicine: Ethical Points to Consider

**DOI:** 10.3390/jpm11060539

**Published:** 2021-06-10

**Authors:** Roel H. P. Wouters, Rieke van der Graaf, Tessel Rigter, Eline M. Bunnik, M. Corrette Ploem, Guido M. W. R. de Wert, Wybo J. Dondorp, Martina C. Cornel, Annelien L. Bredenoord

**Affiliations:** 1Department of Medical Humanities, Julius Center, University Medical Center Utrecht, P.O. Box 85500, 3508 GA Utrecht, The Netherlands; r.h.p.wouters-4@prinsesmaximacentrum.nl (R.H.P.W.); R.vanderGraaf@umcutrecht.nl (R.v.d.G.); 2Princess Maxima Center for Pediatric Oncology, P.O. Box 113, 3720 AC Utrecht, The Netherlands; 3Department of Clinical Genetics, Section Community Genetics and Amsterdam Public Health Research Institute, Amsterdam University Medical Center, Vrije Universiteit, P.O. Box 7057, 1007 MB Amsterdam, The Netherlands; t.rigter@amsterdamumc.nl (T.R.); mc.cornel@amsterdamumc.nl (M.C.C.); 4Department of Medical Ethics, Philosophy and History of Medicine, Erasmus MC, University Medical Centre Rotterdam, P.O. Box 2040, 3000 CA Rotterdam, The Netherlands; e.bunnik@erasmusmc.nl; 5Section Health Law, Department of Social Medicine, Amsterdam University Medical Center, University of Amsterdam, P.O. Box 22660, 1100 DD Amsterdam, The Netherlands; m.c.ploem@amsterdamumc.nl; 6Department of Health, Ethics and Society, Research School CAPHRI for Public Health and Primary Care, Research School GROW for Oncology and Developmental Biology, Maastricht University, P.O. Box 616, 6200 MD Maastricht, The Netherlands; g.dewert@maastrichtuniversity.nl (G.M.W.R.d.W.); w.dondorp@maastrichtuniversity.nl (W.J.D.)

**Keywords:** learning health care, learning healthcare systems, LHS, precision medicine, personalized medicine, genomics, sequencing, ethics, methodology

## Abstract

Learning healthcare systems have recently emerged as a strategy to continuously use experiences and outcomes of clinical care for research purposes in precision medicine. Although it is known that learning healthcare transitions in general raise important ethical challenges, the ethical ramifications of such transitions in the specific context of precision medicine have not extensively been discussed. Here, we describe three levers that institutions can pull to advance learning healthcare systems in precision medicine: (1) changing testing of individual variability (such as genes); (2) changing prescription of treatments on the basis of (genomic) test results; and/or (3) changing the handling of data that link variability and treatment to clinical outcomes. Subsequently, we evaluate how patients can be affected if one of these levers are pulled: (1) patients are tested for different or more factors than before the transformation, (2) patients receive different treatments than before the transformation and/or (3) patients’ data obtained through clinical care are used, or used more extensively, for research purposes. Based on an analysis of the aforementioned mechanisms and how these potentially affect patients, we analyze why learning healthcare systems in precision medicine need a different ethical approach and discuss crucial points to consider regarding this approach.

## 1. Introduction

In recent years, the concept of Learning Healthcare Systems (LHSs) has emerged as a strategy to continuously use experiences in clinical care for quality improvement and research [1]. LHSs aim to systematically study, evaluate and improve quality and efficiency of care while speeding up the process of generating generalizable medical evidence. Proposals to achieve these aims range from optimizing clinical care by learning from routinely collected data to embedding randomized controlled trials (RCTs) in routine clinical care [2].

Roughly during the same time, another concept that promises to revolutionize health care has taken hold: precision medicine, the ambition to tailor disease management to individual characteristics including genomics and lifestyle [3]. Although learning healthcare and precision medicine constitute separate endeavors, these ambitions intersect. To better understand variation within patient groups, ultimately down to the level of individuals, precision medicine can profit from more evidence, from more data, and from a broader range of patients [4]. Yet, traditional forms of research are increasingly considered inadequate at delivering this evidence and involving this broader range of patients. As a result of the decreasing size of patient groups, RCTs in precision medicine are often unable to generate sufficient power. By systematically involving patients in research-oriented activities embedded in routine care, LHSs have the potential to solve this problem of underpowered RCTs [5,6,7]. Contrary to traditional clinical trials, which typically include only a fraction of the entire patient population, learning healthcare studies would bring many more patients into the fold. In addition to enhancing statistical power by increasing sheer numbers, learning healthcare systems can expedite longitudinal follow-up in unselected cohorts, creating opportunities to gather evidence on multidrug combinations, rare side effects, and long-term outcomes of interventions [8,9,10].

Despite its promise to expedite scientific knowledge development and improve clinical outcomes, transitioning to an LHS raises ethical challenges. The ambition to conduct scientific research in conjunction with routine clinical care challenges the traditional ways of ethically and legally evaluating care and research. Medical research activities are typically bound to a different set of ethical, legal, and procedural norms than care activities, with additional requirements for, inter alia, informed consent and ethics oversight. Whether research-oriented activities in LHSs should be governed according to care or research norms remains a debated and contested issue. In addition, it is often unclear how to evaluate learning activities that cannot be classified as research or care but fall somewhere in the middle and, accordingly, it is unclear which norms should apply [11,12]. Additionally, there are concerns as to whether professionals can reconcile their tasks as researchers with clinical obligations toward individual patients. The traditional view is that dual clinician-researcher roles ought to be avoided because patients should be able to trust their doctors as solely acting in the individual patient’s best interest [13]. Clinicians in LHSs also act in the interest of society, serving the common good by advancing research; hence, they may have to balance responsibilities connected to these different roles. Ethicists who came out in favor of learning healthcare transformation have responded to these concerns by pointing out that this transformation can be used as an impetus for revising the incumbent ethical framework. For example, they have posited that learning health care is an opportunity to overcome (unjustified) discrepancies between protections for research participants and patients. Moreover, such a system arguably promotes solidarity and reciprocity because patients, who are beneficiaries of previous scientific work, would get an active role in advancing medical scientific progress [14,15].

While the ethical and legal merits as well as pitfalls of LHSs in general continue to be debated, the fact that learning health care is already gaining traction in precision medicine warrants a more detailed evaluation of ethical aspects in this particular context. In line with the initial 2007 report on the LHS by the US Institute of Medicine, many previous contributors to the debate have focused on comparative effectiveness research involving standard-of-care interventions (e.g., various antihypertensives that are prescribed interchangeably) [1,15]. It is not self-evident that these previously published ethical evaluations of the LHS are equally applicable to LHSs in precision medicine, which is a rapidly evolving and relatively new field. Additionally, the fact that precision medicine as such precipitated myriad ethical debates over the past decade (e.g., regarding genetic discrimination, feedback on incidental findings, and privacy) suggests that learning healthcare transformation in this field calls for additional deliberation. In this paper, we evaluate to what extent precision medicine needs a different ethical approach to combining care and research in an LHS than other contexts for which the LHS has initially been proposed. For the sake of clarity and brevity, this analysis will be focused on specific ethical aspects related to LHS and precision medicine; meanwhile, we fully acknowledge that LHS transformation in general elicits ethical questions that still demand further deliberation. Additionally, we will focus primarily on genomics-guided therapy. Precision medicine is obviously much broader, yet genomics is a key driver of many precision medicine innovations and has been acknowledged as a central feature of LHS transformation in precision medicine [16,17,18].

## 2. The Levers of Learning Healthcare Systems in Precision Medicine

Just as there are many forms of precision medicine, there are many different ways of transforming to an LHS. Instead of one roadmap to learning health care, the literature contains a patchwork of proposals for organizing LHSs, loosely connected by a shared ambition to converge medical practice and research. Some LHSs propose innovative trial or cohort designs that operate entirely within the traditional parameters of research ethics and law, for example in the sense that conducting such research is perfectly compatible with obtaining individual consent. On the other end of the spectrum are proposals to routinely randomize patients between standard-of-care alternatives for treatment on the basis of a general notification that patients are treated in an LHS and may therefore be involved in research-oriented activities without explicit informed consent. With respect to precision medicine, strategies put forward for LHS transformation are equally diverse. It may be helpful to break down the range of options along the lines of the building blocks inherent in the definition of precision medicine, that is, the ambition to take individual (genetic) variability into account in treatment decisions in order to generate better outcomes. From this definition, it follows that there are at least three levers that institutions can pull if they want to implement an LHS. These levers are (1) changing the tests to measure individual variability (such as genes) by implementing additional tests or by widening the scope of tests that are already used, (2) changing the treatments that are prescribed on the basis of the test results and/or (3) changing the handling of data that link variability and treatment to clinical outcomes. Likewise, there are three ways by which patients can be affected if their care is used for research in a learning healthcare system. First, patients are tested for different or more factors than before the transformation. Second, patients receive different treatments than before the transformation. Third, patients’ data obtained through clinical care are used, or used more extensively, for research purposes. We will now analyze these three potential levers of learning healthcare transformation in precision medicine in more detail.

### 2.1. Changing the Scope of the Tests to Measure Individual Variability

The first way in which patients can be affected by an LHS transformation in precision medicine relates to the type and scope of testing that patients receive in clinical care. It has previously been suggested that learning healthcare systems may justifiably add additional tests to clinical workflows for scientific purposes [15]. Applied to precision medicine, this could mean (for instance) that patients whose germline DNA would not have been sequenced outside a learning healthcare system, will now be sequenced as part of this transformation. This resonates with the growing popularity of upfront genome sequencing as part of standard clinical workups, given the increasing clinical applicability and decreasing costs of sequencing [19,20,21,22]. Learning healthcare transformation has been coined as a potential winning strategy to bring genome sequencing to the clinic. Patients who already undergo testing for specific genes may be offered a broader test, such as whole-genome or exome sequencing, or broader analysis of raw sequencing data [23]. The wide range of genomic variants that could thus be revealed would include findings that are not directly relevant or are of unknown relevance to the diagnostic and therapeutic trajectory of the patient, but can nevertheless contribute significantly to research [24]. A joint clinical-research enterprise could justify the necessary investments in infrastructure for a procedure that is thought to be of great promise if undertaken on a large scale but today still offers fairly limited benefits to most individuals [25,26].

Still, using broader tests or broader analyses as a means to pursue research interests in an LHS, even if it aims to improve clinical care in the near future, is not without pitfalls. There is a longstanding consensus that individuals have a legitimate interest in making autonomous decisions as to whether they want to be informed about genetic test results [27,28]. An ethically responsible policy toward genetic or genomic testing enables individuals to make a deliberate choice based on information about what they are tested for and how such tests could impact their lives. This requires dedicated counseling efforts. If large groups of patients are routinely offered genomic testing, it would be quite demanding, if not impossible, for healthcare facilities to live up to these standards in terms of necessary staff and resources [29].

Because of these high standards of information provision and counseling, routinely sequencing genomes of patients without a (direct) clinical need creates ethical challenges that are not typically encountered in other contexts of learning healthcare transformation (e.g., additional routine lab checks). As mentioned above, the argument for learning healthcare systems in general is built on the notion that clinically embedded learning activities should not substantially deviate from the standard-of-care [30]. The case for relatively lenient consent procedures, among other reforms, rests upon the presumption that integrated care-research activities neither impose risks surpassing clinical-level risks (i.e., risks associated with the standard-of-care) nor interfere meaningfully with patients’ values. The practice of testing patients for more genes than required for providing good care has at least the potential to violate both precepts. To protect patients from risks and to honor widely acknowledged standards of autonomous decision-making in genetics, careful additional procedures need to be put in place to navigate LHSs in precision medicine.

### 2.2. Changing the Treatments that Are Prescribed on the Basis of These Tests

The second lever that learning healthcare systems in precision medicine can pull pertains to the prescribing of therapeutics on the basis of genomic profiles and other biomarkers. In the general LHS literature, much attention has been given to embedding (randomized) clinical trials in clinical care. One underlying rationale found in the LHS literature is that many therapeutic choices in clinical care are already quite contingent in the sense that a clinician’s decision to prescribe one particular drug instead of an alternative is oftentimes rooted neither in scientific evidence nor in expert consensus. Sometimes, the clinician may not have a deliberate preference for one drug over the other but makes his or her decision on the basis of habit [31]. This (contested) reasoning stipulates that if prescription behavior is already more or less contingent on arbitrary factors, embedded randomized trials do not pose more risk than patients would have encountered in ordinary clinical care [32].

Notwithstanding the potential of embedded therapeutic research for the future development of precision medicine, therapies in precision medicine are often very distinct from the kinds of drugs that have typically served as case-examples for learning healthcare reform. As explained earlier, paradigmatic cases presented to argue in favor of LHS transformation pertain to standard-of-care interventions that have been used for a long period of time and that have been prescribed more or less interchangeably with comparable interventions. The differences between the latter classes of interventions and those used in precision medicine should not be ignored, as they have potentially far-reaching implications for the ethical justifiability and societal acceptability of learning healthcare systems in precision medicine. A large proportion of precision medicine is aimed either at developing novel therapeutics that are biologically designed to target specific biomarkers or at identifying biomarkers to more accurately select which subsets of patients may benefit from newly designed therapeutics [33,34,35]. Drugs used in precision medicine, and the biomarkers on the basis of which these drugs are prescribed, are oftentimes relatively novel. In addition, in precision medicine, off-label prescription (i.e., prescription for indications not authorized by regulatory agencies), is even more common than in many other fields of medicine [36,37]. As a result, patients who are prescribed precision medicine interventions face different, and comparatively more unknown, risks than patients in the stereotypical contexts for which the LHS concept has originally been developed.

In general, different views exist as to how (and whether) embedded (randomized) comparative effectiveness trials could be institutionalized and under what conditions such trials are justified, particularly in terms of consent and ethics review. For example, some have argued that notifications are sufficient for certain embedded (randomized) comparative effectiveness trials, while others have asserted that explicit consent is still necessary in these contexts (albeit in a potentially different form than the written consent forms that are currently used in research). Yet, even ethicists in favor of learning healthcare reform by and large agree on a key ethical premise underlying the integration of research in clinical care: consent for research can only be replaced by alternatives such as notifications if and only if research activities do not pose substantially more risks than patients would have encountered in ‘ordinary’ clinical care. Hence, for any learning healthcare system to credibly claim that research and care are integrated in an ethically responsible fashion, it is crucial that research-oriented care activities do not exceed this threshold of clinical risks. Otherwise, learning health care is just research by another name, and seamlessly integrated learning activities would just be window-dressing for medical research without consent or appropriate review.

### 2.3. Changing the Handling of Data that Link Variability and Treatment to Outcomes

At present, many blueprints for learning healthcare transformation in precision medicine focus on pursuing generalizable knowledge in an observational manner. The most straightforward examples are registries that continuously extract clinical data from electronic health records and aggregate these data to large datasets, which can subsequently be analyzed for research and quality monitoring purposes. Although this particular type of learning healthcare system does not directly interfere with the clinical journey of a patient, such observational research infrastructures can still affect patients’ outcomes and values (e.g., with respect to privacy). Because of the sensitive nature of (genomic) data being generated and collected in precision medicine, even observational learning healthcare activities are more controversial in this field than in other contexts.

Precision medicine needs precise and comprehensive data to substantiate changes in clinical practice. In genomics-guided precision medicine, this means that data collection ideally comprises entire exome or genome sequences (if available) [38,39]. Subsequently, genomic data need to be linked with broad and granular data encompassing entire clinical trajectories, from patient characteristics via diagnostics and treatment decisions to outcomes [21,40]. All of these factors render precision medicine data more difficult to de-identify and also easier to re-identify, increasing the risk of privacy violations. When such privacy breaches actually occur, the fallout would affect not just individual patients but also their families [41]. Privacy concerns are further increased because building a successful LHS in precision medicine relies on extensive data sharing between institutions (and probably also across borders). Genomic variants correlated with prognosis or treatment outcomes are relatively rare, meaning that clinically relevant conclusions can often only be drawn from databases exceeding the patient population of a single institution [10,42,43].

Apart from the potential impact of systematically collecting genomic and other clinical data for research-related purposes on individuals, one should consider how specific patient populations would be affected by the research elements of LHS transformation. Genomics-guided precision medicine, almost by definition, draws on genetic variances that are not equally distributed among ethnic populations. Certain mutations are more prevalent in, for example, patients of European descent than in other groups [44]. The relative scarcity of genomic data and research in non-white populations has been cited as an important reason to pursue LHS transformation, in order to try to mitigate disparities in precision medicine [8]. However, it has also been cautioned that genomics data can be used for research connected to race and ethnicity in ways that may give rise to group-level discrimination and stigmatization even when these data cannot be traced back to the individuals to whom these genomic sequences belong [45]. Hence, routine inclusion of patients’ genomic data in research databases would elicit more controversy and warrants more extensive ethical reflection than other types of LHS research.

## 3. Discussion

The analysis of the abovementioned elements shows that LHS transformation as a model for innovation in precision medicine encompasses opportunities but also challenges. Understandably, the prospect of a continuous cycle of (scientific) evaluation of care, rapid adaptation of clinical strategies, and subsequent re-evaluation of those strategies has been heralded as a great opportunity for a rapidly developing field like precision medicine. Stakeholders have noted the potential of systematic clinical data collection not only for precision medicine research but also to provide patients with up-to-date, evidence-based clinical advice (optimally tailored to their individual situations). Yet, the challenges related to precision medicine LHSs may well exceed the challenges posed by earlier examples of LHS reform described in the literature. This conclusion has important implications for how to develop ethically acceptable precision medicine LHSs. We will now proceed to discuss three major strategies at hand for managing such challenges and building LHSs in precision medicine in an ethically sound way: consent, independent review and public accountability.

### 3.1. Consent

Consent reform is often considered to be part and parcel of LHS transformation. Streamlining consent requirements across the domains of care and research would contribute to making scientific learning a seamless component of health care delivery [10,46]. Alternatives to traditional informed consent for medical research include amendments to form (e.g., from written to oral consent, and from written consent forms to videos), substance (from specific to broad consent) or default (from opt-in to opt-out) [38]. Arguably the most far-reaching (and ethically most controversial and problematic) alternative would be to replace consent with general notifications, meaning that patients would be notified that their healthcare practice is a learning healthcare system and has therefore embedded research-oriented activities into clinical routines [47,48].

Proponents of consent mitigations have defended their proposals by pointing out that the care received by patients in envisioned LHSs does not differ from normal care in any meaningful way. For this reason, mitigating informed consent requirements in the type of contexts for which the LHS was originally proposed allegedly does not deprive patients of any relevant freedom to make meaningful choices. For example, these authors presume that patients would not have a strong interest in being able to choose between two drugs that, from a medical perspective, are more or less interchangeable [14]. In precision medicine LHSs, this presumption is problematic. Patients would undergo tests or receive relatively novel interventions that they would not otherwise have encountered in daily clinical practice, potentially exposing patients to relatively high levels of risk. Therefore, precision medicine does not seem to be an ideal setting to experiment with consent reform, particularly with highly controversial alternatives such as replacing individual consent with notifications [14].

Healthcare leaders aiming to apply learning healthcare transformation to precision medicine should seek new opportunities for scientific evidence-gathering within the parameters of patients providing explicit authorization for their participation in research. Literature provides a range of options, including broad consent and “thick” opt-out consent systems, that may be justifiable alternatives to traditional (i.e., specific, opt-in) consent for certain applications of the LHS in the context of precision medicine [49]. Pilot projects are needed to determine whether these theoretically well-established alternatives are indeed feasible, reliable and acceptable strategies to obtain patients’ approval in LHSs in precision medicine. Alternative forms of consent should also be in accordance with national and international law.

### 3.2. Oversight

The rise of combined care-research activities necessitates reflection on the role of research ethics committees (RECs) in this process. Traditionally, activities aimed at creating generalizable medical knowledge are labeled as research, which triggers regulatory mechanisms leading to independent review by RECs. If care and research become more entangled, the criteria formerly used to prompt review will no longer suffice to discern which activities need to be assessed. This poses a dilemma with respect to the mandate of RECs.

Intertwinement of care and research goals, sometimes within the same activity, underscores the need for deliberated assessment by an independent (ethics) committee. This oversight need not necessarily be identical to current oversight procedures for biomedical research involving humans by means of RECs; novel oversight mechanisms for hybrid activities have already been suggested, such as a “multifaceted system of research review and oversight” [50,51]. Disentangling to what extent risks are rooted in research-oriented parts of the activity, and subsequently how these risks should be weighed against potential benefits, requires careful judgment and often discussion between committee members. In other words, the rise of hybrid activities in an LHS would probably increase the need for oversight by RECs or similar organizational structures.

At the same time, subjecting all combined care-research activities to independent review might undermine the whole project of more seamless integration between care and research. The burden for RECs and professionals may increase substantially, which might slow down rather than speed up scientific learning in health care. In addition, it is questionable as to whether RECs have the capacity to properly evaluate activities that currently fall outside of their scope and mandate [52,53]. In sum, a rethinking of the entire system of ethics oversight is needed when moving toward a learning healthcare system, particularly in precision medicine.

One pertinent example of how the scope of ethics oversight may need to be expanded is constituted by the management of genomic data safety. It is obvious that data protection demands technical ingenuity to develop innovative solutions in order to protect the privacy of individuals in the genomic era, especially if individuals are increasingly treated as patients and participants at the same time. The technical and logistical aspects of the necessary data protection approaches have been discussed extensively elsewhere [54]. Protecting individuals’ intricate interests related to genomic sequencing data demands more than just technical security measures; it also requires solid organizational embeddedness of data protection in existing or novel ethics oversight mechanisms. While the trap of engaging in genetic exceptionalism should be avoided, i.e., genetic data should generally be treated as qualitatively rather than categorically different from any other type of data [55], the intricate features of genomic sequencing data, (more easily identifiable, more stable over time and at-least perceived as more personal than many other data) demands development of careful oversight mechanisms in the face of learning healthcare transformations. Ideally, assessing the most appropriate level of genomic data protection at the intersection of care and research would be an integrative part of ethics oversight. Currently, it is not uncommon that the supervision of data protection is vested in data management boards that operate separately from research ethics committees [56]. Instead of treating data protection as a separate matter, however, handling and management of genomic data should take central stage in ethically responsible governance. Recognizing that the use of genomic data may alternate between primarily care- or research-centered, the ethical and legal scaffold should also be adapted to make sure that individuals are adequately protected from harms due to unjustified uses of their data [57]. Moreover, adequate ethics oversight and governance of genomic data requires a longitudinal approach that is not merely concerned with up-front assessment of benefits and risks (i.e., before the study commences), but extends to offering optimal protection even up to managing the fallout of privacy breaches and hacks [57]. In the EU, this broader estimation of data protection risks is also what the General Data Protection Regulation (2016/679) expects from what is known as a ‘data protection impact assessment’, provided for in Article 35 of the Regulation [58].

### 3.3. Public Engagement and Accountability

A primary concern regarding LHS transformation in precision medicine on a societal level is that care–research integration in precision medicine will disproportionally disadvantage underprivileged communities. On the one hand, as said, there is a well-described underrepresentation of patients of non-European descent in precision medicine data [59]. If LHS transformation develops faster in relatively white and affluent regions, this may increase rather than mitigate the disparities in research data between populations [60,61].

On the other hand, racial minorities tend to be more reluctant toward participating in genomics research and non-white patients seem more frequently opposed to research without explicit consent, also with regards to scientific learning activities in an LHS [44,62]. As a result, embedding scientific research in clinical care could deter non-white populations from seeking precision medicine-related care in newly transformed LHSs. Besides the potential effects on access to health care, this would render LHS transformation ineffective at filling current data gaps between white and non-white populations [39].

It is unlikely that these challenges can be resolved through consent and oversight-oriented measures alone. Issues related to representation and diversity in precision medicine transcend the level of individual healthcare organizations and are intertwined with a broader range of societal dynamics [63]. Still, this intertwinement does not waive the responsibilities of organizations to address these issues in the process of transitioning to an LHS. For this purpose, healthcare organizations also need to look beyond consent and oversight measures centered around individual patients. In order to accomplish inclusive and equitable LHS policies, strategies aimed at improving accountability and engagement with specific communities as well as the general public need to be developed.

## 4. Conclusions

LHS transformation is a promising strategy to bring innovations in precision medicine, such as genomics-guided therapy, to daily clinical practice. LHSs can provide a platform for continuous scientific evaluation of precision medicine innovations when these find their way into the clinic. This way, LHS development can be conducive to accomplishing evidence-based precision medicine. Unsurprisingly, the LHS concept has drawn considerable attention from the precision medicine community. By offering an approach to combine care and research, learning health care may provide solutions for some of precision medicine’s central needs. Yet, several aspects of learning health care in precision medicine render the transformation (even) more ethically controversial than in other contexts. The potential impact on individuals’ and communities’ interests ought to be given due consideration and needs to be managed accordingly.

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
