# Peer review of "Towards a Responsible Transition to Learning Healthcare Systems in Precision Medicine: Ethical Points to Consider"

_jpm, 2021, doi:10.3390/jpm11060539_

Round 1

Reviewer 1 Report

I had the pleasure of reviewing the article entitled: “Towards a responsible transition to Learning Healthcare Systems in precision medicine: ethical points to consider”. The article is very interesting, well written and easy to read. The introduction is clearly written, the whole is supported by properly selected literature. It is exhaustive in each of its sections and deals with a very interesting topic. I consider the article important from the medical point of view and worth reading and attention. In my opinion the paper is suitable to be published in Journal of Personalized Medicine.

Author Response

Thank you for your time and consideration. We are very pleased with your positive reception of our manuscript.

Reviewer 2 Report

Well developed and clearly articulated approach using 3 levers to characterize the differences between conventional LHS and precision medicine LHS to identify unique challenges and associated ethical issues to support a conclusion and needs for making them ethically acceptable including how they are not yet adequately addressed by 3 major strategies (consent, independent review, and public accountability), and can further contribute to inequities/disparities for minority ethnic and disadvantaged populaitons.

Author Response

(The authors gave the same response as above.)

Reviewer 3 Report

This paper contributes an important and timely discussion on learning healthcare system in precision medicine (from ethical perspective) which can be significant in transforming the way of delivering clinical care and improving patient health.  The paper is well organized and written, I enjoyed reading it. 

My only major comment is: the paper did intensive considerations and well laid out potential concerns and challenges regards to the sensitiveness of genomic data including identification, privacy and larger concern of group-level discrimination, stigmatization to a certain race/ethnicity group. First of all, I totally agree genomic data elicit more controversy and warrants more extensive ethical reflection. And I even think there are other potential concerns which can be even worse, e.g. genomic data being used to initiate war against a certain race/ethnicity population. Second, even the authors had intensive discussion on potential concerns and challenges as referred above, the discussion on potential solutions to address and prevent these concerns seem fall short, i.e. what we can do to warrant security of genomic data and prevent inappropriate use of it. I would appreciate more discussion on this regard as I believe that is one of the most important parts in integrating precision health in clinical care and learning health system.   

Author Response

Thank you for reading our manuscript carefully and we are very happy that the manuscript was well-received. With regards to your concern about the potential of genomic data to wage a war against racial groups, we agree that the potential of these data to be misused to discriminate against, or otherwise harm, racial groups, is currently underestimated by the medical community. For this reason, the manuscript already emphasized the need for more debate on how the use of genomic data in precision medicine-directed learning healthcare systems could cause discrimination and stigmatization and how these could be prevented (e.g. revised manuscript lines 404-413). This discussion in the manuscript mainly revolves around harms that may be foreseeable but nevertheless are unintended. Your comment seems to suggest that the medical and scientific community should also anticipate deliberate efforts to compromise the interests of racial groups through collection of genomic data. This is an interesting observation that certainly warrants further study. Yet, we do feel that analysis and evaluation of this concern would be deserving of a separate paper, or even multiple paper, particularly because this topic is so controversial (and consequential). Also, the potential ramifications of this observation pertain to genomic research in general and extend way beyond the sphere of learning healthcare transformations in precision medicine that is the focus of our current manuscript. We therefore respectfully decided not to dedicate an additional section on this topic in our manuscript, but your comment is definitively food for further thought and deliberation.

Data protection is a major topic in genomics research that has received ample attention in the literature, which is why we had not dedicated more space to this topic in our original manuscript. However, we fully agree that data protection is a vital part of any application of genomics in clinical care and research. Also, we agree that learning healthcare transformation makes this issue even more urgent. Therefore, we added an entire paragraph to the discussion section in which we discuss the importance of data protection and its relation to what we already recommended with regards to ethics oversight. Please find this additional section in lines 357-387 in the revised manuscript.